# Exposing *Salmonella* Senftenberg and *Escherichia coli* Strains Isolated from Poultry Farms to Formaldehyde and Lingonberry Extract at Low Concentrations

**DOI:** 10.3390/ijms241914579

**Published:** 2023-09-26

**Authors:** Irena Choroszy-Król, Bożena Futoma-Kołoch, Klaudia Kuźnik, Dorota Wojnicz, Dorota Tichaczek-Goska, Magdalena Frej-Mądrzak, Agnieszka Jama-Kmiecik, Jolanta Sarowska

**Affiliations:** 1Department of Basic Sciences, Faculty of Health Sciences, Wrocław Medical University, Chałubińskiego 4, 50-368 Wroclaw, Poland; irena.choroszy-krol@umw.edu.pl (I.C.-K.); magdalena.frej-madrzak@umw.edu.pl (M.F.-M.); agnieszka.jama-kmiecik@umw.edu.pl (A.J.-K.); jolanta.sarowska@umw.edu.pl (J.S.); 2Department of Microbiology, Faculty of Biological Sciences, University of Wrocław, Przybyszewskiego 63–77, 51-148 Wroclaw, Poland; klaudia.maria.kuznik@gmail.com; 3Department of Biology and Medical Parasitology, Faculty of Medicine, Wrocław Medical University, Mikulicza-Radeckiego 9, 50-345 Wroclaw, Poland; dorota.wojnicz@umw.edu.pl (D.W.); dorota.tichaczek-goska@umw.edu.pl (D.T.-G.)

**Keywords:** *Salmonella* sv. Senftenberg, *Escherichia coli*, biofilm eradication, inhibition, formaldehyde, *Vaccinium vitis-idaea*

## Abstract

European Union (EU) countries strive to improve the quality and safety of food of animal origin. Food production depends on a good microbiological quality of fodder. However, feed can be a reservoir or vector of pathogenic microorganisms, including *Salmonella* or *Escherichia coli* bacteria. *Salmonella* spp. and *E. coli* are the two most important food-borne pathogens of public health concern. Contamination with these pathogens, mainly in the poultry sector, can lead to serious food-borne diseases. Both microorganisms can form biofilms on abiotic and biotic surfaces. The cells that form biofilms are less sensitive to disinfectants, which in turn makes it difficult to eliminate them from various surfaces. Because the usage of formaldehyde in animal feed is prohibited in European countries, the replacement of this antibacterial with natural plant products seems very promising. This study aimed to assess the inhibitory effectiveness of *Vaccinium vitis-idaea* extract against biofilm produced by model *Salmonella enterica* and *E. coli* strains. We found that formaldehyde could effectively kill both species of bacterial cells in biofilm, while the lingonberry extract showed some antibiofilm effect on *S. enterica* serovar Senftenberg. In conclusion, finding natural plant products that are effective against biofilms formed by Gram-negative bacteria is still challenging.

## 1. Introduction

Food is a rich source of nutrients necessary for the proper functioning of the body. Unfortunately, it is also the optimal living environment for various microorganisms, including bacterial pathogens that obtain nutrients necessary for their development from the inhabited food. Microorganisms can enter food from many sources, including water, raw products, and animals, and can persist on a variety of surfaces [1]. Food products can become contaminated at any stage of the production cycle, despite the implementation of all required cleaning and disinfection protocols. Moreover, the spread of infectious disease outbreaks is always associated with significant economic losses [2]. According to reports from the European Food Safety Authority (EFSA) and the European Center for Disease Prevention and Control (ECDC), salmonellosis is one of the most common food-borne zoonoses in the countries of the European Economic Area (EEA) and the European Union (EU), and the number of confirmed cases in 2020 was 52,702 [3]. In Poland, in 2020, *Salmonella* rods were responsible for over 87.5% of food poisoning cases [4]. *Salmonella* infections are most often caused by the consumption of contaminated dairy products, meat, eggs, as well as fruit and vegetables, which may be the result of contamination of the soil or water used in the cultivation [5]. In the USA, *Salmonella enterica* serovar Senftenberg is among the top five serovars isolated from food [6], and there is little information about its susceptibility to chemical disinfectants. Colibacillosis, the etiological factor of which is pathogenic *Escherichia coli* strains, is one of the main causes of mortality and morbidity in poultry. In the US, it has been estimated that economic losses to the broiler industry can be as high as US $40 million per year. Pathogenic avian strains of *E. coli* can be transmitted to humans through contaminated poultry meat and direct contact with birds or their feces [7]. It is worrying that the number of reports of poultry products contaminated with *Salmonella* bacteria has increased in recent years. This may be partially due to the prohibition of formaldehyde usage to disinfect poultry feed in the EU. This fact forces both poultry farmers and food producers to take more radical steps to effectively eliminate this pathogen from food products that reach consumers [4]. In addition, as in the case of humans, the phenomenon of asymptomatic carriage of *Salmonella* has also been confirmed in animals, which significantly contributes to the spread of these pathogens in the environment. Soil contaminated with excrement also plays an important role in the transmission of *Salmonella*, being a potential source of infection for both humans and animals. The consequence of the high survival rate of these rods in the soil is the possibility of contamination of crops that can be eaten by humans or used in poultry farms as litter [8,9]. In the food industry, a significant threat is the contamination of surfaces, raw materials, or products with microorganisms capable of forming a biofilm [5,10]. Bacterial biofilms are detected on countertops and elements of processing devices that are made of materials such as nylon, stainless steel, glass, rubber, aluminum, polystyrene, and plastic [2,11]. In poultry processing, protection systems and chemicals are consistently applied to minimize the risk of contamination. However, numerous studies have shown a high prevalence of *Salmonella* in environmental samples collected after cleaning and disinfection of both laying hens and broiler houses, indicating their ineffectiveness against these bacteria [12]. Bacteria embedded in the extracellular polymeric substances (EPS), consisting of a mixture of polymeric compounds such as proteins, polysaccharides, lipids, and nucleic acids, are nearby and form channels for the transport of nutrients, oxygen, and water [13]. It is believed that the most important factor contributing to the reduced sensitivity of bacterial biofilms to bioactive substances is the specific structure of the biofilms [5]. Cells that exist in a biofilm, compared to planktonic cells, have 10 to 1000 times higher resistance to biocides [14]. By far, the best strategy to eliminate bacterial biofilms from food processing environments is to prevent their formation [2]. The elimination of most risks related to contamination with pathogenic bacteria is achieved by proper cleaning and disinfection processes. The substances used in the disinfection process may be bacteriostatic or bactericidal [15]. Chemicals currently used in disinfection processes include, among others, alcohols, acid compounds, aldehyde-based agents, quaternary ammonium compounds, hydrogen peroxide, ozone, peracetic acid, phenols, and surfactants [16]. Commercial disinfectants contain one chemical compound or their mixture, which is usually recommended due to the increase in the spectrum of action and the effectiveness of the disinfectant [17,18]. The EU Commission Implementing Regulation 2020/1763 of 25 November 2020 approved formaldehyde as an active substance for use in biocidal products belonging to the PT2 group (private and public health disinfectants and other biocidal products) and the PT3 group (biocides intended for the maintenance of veterinary hygiene), subject to compliance with the specifications and conditions set out in the annex to this regulation [19,20]. Due to the increasing frequency of detection of bacterial strains forming biofilms resistant to disinfectants and antibiotics, it is important to develop alternative strategies for their elimination. In the era of growing resistance to antibiotics and disinfectants among bacteria, alternative antibacterial agents are sought. The studies conducted so far on the antibacterial activity of plant extracts indicate that they, and compounds isolated from them, can be successfully used to inactivate microorganisms. The antibacterial effect of the extracts may be related to the modification of bacterial surface structures, inhibition of the synthesis of nucleic acids, and enzymatic and structural proteins [21,22]. Lingonberry is a medicinal plant whose leaves and fruits are valuable pharmaceutical raw materials because of their phenolic constituents. Lingonberry leaves are used to treat diseases of the urinary system, and they also exhibit antidiarrheal or antioxidant activity [23,24,25,26,27].

Studies on the composition of *Vaccinium vitis-idaea* (lingonberry) extract showed the presence of proanthocyanidins, hydroxycinnamic acid (HCA) derivatives, flavonols, anthocyanins, and benzoylglucose. Proanthocyanidins appear as the main component of the fruit, although other quantitative studies have shown the presence of A-type and B-type proanthocyanidins or anthocyanins, HCA, and flavonols [23]. A-type proanthocyanidins have been attributed with antiadhesive properties against *E. coli* bacteria, and thus have a protective effect on the urinary tract [28]. Polyphenols have shown antimicrobial properties against inter alia *Salmonella enterica* sv. Typhimurium, antiaggregation against *Streptococcus mutans* with *Fusobacterium nucleatum*, and antiadhesion against *Neisseria meningitidis* or oral streptococci in biofilm formation, among others [29]. The antimicrobial effect of lingonberry is attributed to the abundant flavonols, quinones, and flavonoids present in its composition. These substances have lipophilic properties, based on which they cause damage to the cytoplasmic membrane and the cell wall of microorganisms. In addition, they inhibit the synthesis of nucleic acids and enzymatic and structural proteins [21]. It was confirmed that flavonoids such as kaempferol, naringenin, quercetin, and apigenin were associated with a decrease in biofilm synthesis by disrupting quorum sensing. The effects of the lingonberry extract obtained from plants collected in different geographical zones differed. There are significant differences in the content and profile of phenols isolated from lingonberries depending on the region in which they grow and the cultivation, variety, ripening stage, environment, soil conditions, weather, or extraction methods [30]. It was shown that the solubility of phenolic compounds is higher in alcohols, therefore, in the case of the ethanol–water extract, a higher content of these compounds was obtained compared with the aqueous extract. The total content of flavonoids in wild lingonberry collected in Poland ranged from 522 to 647 µmol/100 g for the ethanol-water extract and 255 to 353 µmol/100 g for the water extract [31].

Both *Salmonella* strains and *E. coli* are huge problems in the food industry, especially the poultry industry. Our work includes preliminary studies on the biofilm formation capacity of *Salmonella enterica* subsp. *enterica* serovar Senftenberg and *E. coli* strains. Eradication of the biofilms using formaldehyde and inhibition of biofilm formation with *V. vitis-idaea* extract were performed. These studies are a reference point for testing the antibacterial properties of a wide range of natural substances, such as plant extracts, believed to be safer than chemical agents.

## 2. Results

### 2.1. Assessment of Biofilm Formation by Salmonella Senftenberg and Escherichia coli Strains

The biofilm formation capacity of the tested bacterial strains was determined in polystyrene microtiter plates. The intensity of biofilm formation was assessed based on spectrophotometric absorbance (OD) readings. It was shown that there were differences in the intensity of biofilm formation among the tested strains (Figure 1). The OD values for the *S.* Senftenberg strains ranged from 0.129 to 0.234 (±0.007–0.076). *E. coli* strains are better producers of biofilm (especially WW01, CJ01) as evidenced by their higher OD values of 0.161–0.367 (±0.022–0.056).

### 2.2. Evaluation of the Effectiveness of Formaldehyde in the Eradication of Biofilms Formed by Salmonella Senftenberg and Escherichia coli Strains

Formaldehyde is known as a very effective antibacterial substance, but there is no information about its activity against biofilms formed by *Salmonella* Senftenberg and only a little about *E. coli* rods. The results of the tests performed for tested strains are summarized in Table 1 and Table 2. The results presented in Table 1 indicate that 2.0% of formaldehyde completely eradicated the biofilm formed by *Salmonella* Senftenberg strains after a 1 min of exposure. In the case of lower concentrations of formaldehyde (0.2% and 0.02%), only a partial antibiofilm effect of this compound was noted, as has also been presented in Appendix A.

In the case of four *E. coli* strains, WW02, KA01, CM02, and DP01, it was shown that 2.0% formaldehyde completely reduced the biofilm after just 1 min of action. The ineffectiveness of the disinfectant after 1 min was demonstrated for two strains, WW01 and CJ01, which were found to be the top biofilm producers among tested isolates (Figure 1). The usage of 2.0% formaldehyde resulted in the complete eradication of the biofilms after 15 min of exposure (Table 2). Based on the data presented in Appendix A, formaldehyde in concentrations of 0.2% and 0.02% inhibited the biofilms formed by *E. coli* strains much less significantly at both lengths of time used (1 and 15 min).

### 2.3. The Influence of Formaldehyde on the Survival of S. Senftenberg and E. coli Strains in Biofilms

This experiment aimed to show the survival percentage in biofilms, assessed using the BOAT method, which is described in the previous paragraph. Regarding the percentage of bacterial survival in the formed biofilms, the following criteria have been adopted:survival 0.0–5.0%—total inhibitionsurvival 5.1–25.0%—slight inhibitionsurvival 25.1–75.0%—moderate inhibitionsurvival 75.1–95.0%—strong inhibitionsurvival 95.1–100%—no inhibition

Based on the data in the previously mentioned Appendix A, it was found that formaldehyde at a concentration of 2.0% showed a bactericidal effect on all tested strains of *S.* Senftenberg, reducing the viability of bacteria to zero after only 1 min of action. Lower concentrations of formaldehyde (0.2%, 0.02%) significantly reduced the cell number of most biofilm-forming *S.* Senftenberg strains. It is worth emphasizing that this effect was dependent on the exposure time of the rods to the disinfectant and the tested bacterial strain. Formaldehyde at a concentration of 0.02% (Figure 2A), although the least effective, significantly reduced the percentage of viable cells in three out of five analyzed strains of *S.* Senftenberg after 1 min of activity (Appendix A). Extending the time of its action to 15 min caused only moderate or slight inhibition of bacterial growth in comparison to the controls, but all the results were statistically significant (*p* ≤ 0.05). Unfortunately, in two cases (*S.* Senftenberg 132, 135), the prolongation of the formaldehyde activity time to 15 min did not change the number of viable cells in comparison to the 1 min exposure (*p* > 0.05). As can be seen in Figure 2B and Appendix A, formaldehyde at a concentration of 0.2%, after 1 min of activity, had a moderate reduction rate in terms of the survival of the *S.* Senftenberg strains (*p* ≤ 0.05). Extending the exposure time of bacteria to this disinfectant to 15 min resulted in moderate (*S.* Senftenberg 131, 132, 133) or strong (*S.* Senftenberg 134, 135), reduction in the survival of rods in the formed biofilms (*p* ≤ 0.05). The highest concentration of formaldehyde used (2%) completely inhibited the growth of the strains and, thus, the formation of biofilm after only 1 min of exposure. Formaldehyde concentrations that were 10 times and 100 times lower also had a disinfecting effect, significantly reducing the survival of the tested strains after 15 min of exposure. Thus, it was clear that the antibiofilm effect of formaldehyde was dependent not only on the concentration, but also on the time of its action.

An in-depth analysis of the results presented in Figure 3A and Appendix A shows that formaldehyde used in the lowest concentration (0.02%) for 1 min significantly reduced the number of viable cells to 41% survival (WW01), 59% survival (KA01), 75% survival (CM02), and 77% survival (CJ01). Extending the duration of the disinfectant action to 15 min resulted in a significant reduction in the number of living cells of six *E. coli* strains. The results on to the action of 0.2% formaldehyde presented in Figure 3B show that a 1 min exposure of *E. coli* strains to 0.2% formaldehyde resulted in average growth inhibition (*p* ≤ 0.05), except for the *E. coli* WW02 strain, for which no inhibition of survival was observed. Extending the exposure time of 0.2% formaldehyde to 15 min significantly reduced the number of bacteria of all *E. coli* strains except CM02, whose survival was almost the same as after 1 min of exposure. Interestingly, after the 15 min treatment with the disinfectant, the growth of *E. coli* CJ01 was almost completely inhibited (4% survival).

### 2.4. Antibiofilm Properties of V. vitis-idaea Extract in Biofilms Formed by S. Senftenberg and E. coli

The antibiofilm properties of the lingonberry extract were tested against strains that were regarded as the most intensively biofilm-forming: *S.* Senftenberg 131 and 132 and *E. coli* WW01 and CJ01. It was hypothesized that lingonberry extract might possess antibiofilm properties against the tested rods. If so, inhibition of biofilm formation would be observable when compared with the controls. It can be noted that the plant extract at all concentrations inhibited the biofilm formation by both *Salmonella* strains, except for a situation where *S.* Senftenberg 132 was treated with extract in the concentration of 2.0 mg/mL (*p* > 0.05) (Figure 4A). In the case of the *E. coli* CJ01 strain, four extract concentrations (0.02–2.0 mg/mL) significantly reduced its antibiofilm abilities (*p* ≤ 0.05) (Figure 4B). Interestingly, in the case of *E. coli* WW01, only low concentrations in the range of 0.005–0.02 mg/mL exhibited antibiofilm activity (*p* ≤ 0.05) (Figure 4B). It seems that the usage of plant extract, even in concentrations considered high, may be insufficient for the effective elimination of bacteria existing in consortia.

## 3. Discussion

Effective pollution control in the broad area of the food industry has in recent years become one of the main areas of interest for scientists [15,19]. One of the priorities in the field of food microbiology is searching for ways to effectively eradicate the bacterial biofilm that is the cause of the growing resistance to disinfectants and antibiotics [11,14]. Colibacillosis and salmonellosis are examples of threats to animal production, especially in the poultry industry [7,8]. Therefore, our research focused on strains of *E. coli* isolated from chicken feces and *S.* Senftenberg strains isolated from poultry feed. In the study by Dittoe et al. [32], aqueous solutions of formaldehyde in concentrations of 0.05, 0.1, 0.12, and 0.2% were used as additives to the protein feed material, which was a meat-and-bone meal. The feeds were contaminated with the *Salmonella* Infantis marker strain. The results showed that only formulations containing formaldehyde at a concentration greater than 0.1% were effective in preventing the growth of *S.* Infantis. Poultry application and antibacterial mechanisms were extensively described in the review by Ricke et al. [33]. Cochrane et al. [34] compared the effectiveness of four different feed additives: a commercial preparation (containing 0.3% formaldehyde and 2.0% mixture of medium-chain fatty acids (MCFA)), 2.0% mixture of essential oils, 3.0% mixture of organic acids, and 1.0% sodium bisulfate. The feeds containing the above-mentioned additives were then inoculated with *Salmonella* Typhimurium. The most effective chemical additive for inhibiting the growth of rods was a commercial formulation including formaldehyde and MCFA. In the case of disinfectants, it is also recommended to use additives containing various antimicrobial compounds to minimize the generation of cross-resistance among bacteria. Some commercial formaldehyde-based products also contain acids, such as propionic acid. The combination of these two compounds results in a synergy of action, which in turn allows for the use of lower concentrations of formaldehyde and acids [35]. The disinfection process also plays an important role in the control of bacterial infections. Gosling et al. [36] investigated the effectiveness of disinfectants in the eradication of *Salmonella* in animal husbandry facilities. The disinfectants contained active compounds such as chlorocresol, iodine, peracetic acid, glutaraldehyde/formaldehyde, or glutaraldehyde/quaternary ammonium compounds (QAC), as well as potassium peroxymonosulfate. The studies assessed the stability of the disinfectants when used over time and their ability to eliminate *Salmonella* Typhimurium from the biofilm mass. It was found that the preparations containing chlorocresol and glutaraldehyde/formaldehyde were the most effective in eliminating these bacilli from biofilms. The current study aimed to check the effectiveness of formaldehyde as a disinfectant used, inter alia, in the poultry industry against strains of *S.* Senftenberg isolated from poultry feed and strains of *E. coli* isolated from chicken excrements. The obtained results indicate that in the case of all tested strains of *Salmonella* Senftenberg, the bactericidal concentration of formaldehyde, which effectively prevented the regeneration of biofilm, was 2.0%. However, it is worth noting that the lower concentrations of formaldehyde (0.2% and 0.02%) also had a good antibacterial effect. Although they did not kill all *S.* Senftenberg and *E. coli* cells, they significantly reduced the number of bacteria in biofilms. At this point, another very important aspect should be touched upon. In industrial conditions as well as in animal husbandry rooms, the disinfection process is preceded by the removal of organic matter and washing. For the disinfectant to effectively remove the biofilm present on a given surface, it is necessary to ensure optimal conditions for its operation. This is achieved by thoroughly drying the surfaces, especially in gaps or in places where different surfaces are joined together [16,19]. Water accumulates in inaccurately drained places, which may cause the dilution of the applied preparation, which results in the exposure of bacterial cells to sub-lethal concentrations of the disinfectant [15]. Many authors indicate the ability of microorganisms to adapt to sub-inhibitory concentrations of disinfectants, which may result in an increase in resistance to a given chemical compound [37,38,39,40]. The results obtained by Futoma-Kołoch et al. [41] confirmed that the exposure of five strains of *S.* Senftenberg, the same as in this study, to increasing concentrations of a commercial biocide containing triamine, 2-aminoethanol, cationic surfactants, nonionic surfactants, and potassium carbonate, did not produce antibiotic-resistant variants. Cross-resistance has also been reported among pathogens associated with the food industry. This phenomenon is characterized by the formation of resistance mechanisms by cells to a specific substance, which at the same time affects the partial or complete insensitivity to other substances belonging to the same or another class of antibacterial compounds [15]. It is disturbing that the exposure of *Salmonella* cells to an agent containing, inter alia, formaldehyde resulted in the development of stable cell variants overexpressing the efflux AcrAB-TolC pumps and variants showing reduced sensitivity to tetracycline, ciprofloxacin, ampicillin, and chloramphenicol compared with wild-type cells [42]. Additionally, TolC protein is essential for *S.* Typhimurium to achieve poultry colonization [43]. In a study by Karatzas et al. [44], the wild strain *S.* Enterica serovar Typhimurium SL1344 and its three variants (OXCR1, QACFGR2, TOPR2), each obtained after sequential incubation in a sublethal concentration of a mixture of chemicals (OXC—oxidizing compounds containing inorganic peroxygen compounds, inorganic salts, organic acid, an anionic detergent, fragrance, and dye; QACFG—a dairy quaternary ammonium sterilizer containing nonionic surfactant and excipients; and TOP—a biocide composed of tar acids, organic acids, and surfactants), were then treated with quaternary disinfectants (ammonium salts, formaldehyde, glutaraldehyde, and triclosan). Unfortunately, these treatments led to increased levels of resistance to ciprofloxacin, chloramphenicol, tetracycline, and ampicillin. Detailed biochemical and genetic analysis of the tested strains showed the overexpression of several different proteins that protect bacteria against oxidants, peroxides, and disulfides. In addition, a decreased expression of outer membrane proteins and overexpression of AcrAB-TolC pumps were observed, which was also accompanied by a reduction in the invasiveness of these microorganisms [44]. In research conducted by Oosterik et al. [45], 97 *E. coli* strains, isolated from laying hens, were analyzed in terms of their resistance to antibiotics and disinfectants commonly used in the poultry industry (formaldehyde, glyoxal, glutaraldehyde, quaternary ammonium compound, and hydrogen peroxide). Resistance to ampicillin (35.1%), sulfonamides (41.2%), nalidixic acid (38.1%), and tetracycline (53.6%) were demonstrated among the tested strains. At the same time, no phenotypic resistance to the above-mentioned disinfectants was demonstrated in the conducted studies. Our research also observed that the multi-drug-resistant *E. coli* CM02 strain did not show resistance to 2.0% formaldehyde. Another study by Romeu et al. [46] investigated the effect of sub-lethal concentrations of various disinfectants (benzalkonium chloride, sodium hypochlorite, and hydrogen peroxide) on the biofilm formation capacity of *Salmonella* Enteritidis and changes in the susceptibility of these bacteria to antibiotics. Exposure to disinfectants did not change the sensitivity of *Salmonella* to antibiotics. As a result of exposure to sodium hypochlorite and hydrogen peroxide, the reference strain *Salmonella* Enteritidis NCTC 13,349 increased its ability to create biofilm, which may indicate that the cells of these bacteria, under the influence of disinfectants, over-expressed genes for virulence and stress response factors. The study by Course et al. [12] determined how poultry house cleaning and disinfection procedures recommended for broiler producers affect the survival of *S.* Enterica and *E. coli*. Studies have shown that dry cleaning was more effective in controlling *S.* Enterica, while wet disinfecting better reduced *E. coli* cells. Analyzing our own results on the survival of *S.* Senftenberg and *E. coli* strains depending on the duration of formaldehyde action, it was found that formaldehyde at a concentration of 0.2% was always more effective after 15 min. A similar relationship was also observed in 7 out of 11 tested strains, when exposed to 0.02% formaldehyde. The problem of the appearance of bacteria capable of detoxifying formaldehyde is discussed in the literature [47]. These include *Neisseria meningitidis*, *Haemophilus influenzae*, *Bacillus subtilis*, and *E. coli*. An archetypal defense against formaldehyde toxicity is its oxidation to non-toxic formate via glutathione-dependent or -independent dehydrogenases. The development of bacterial biofilms on commercial surfaces in food production and processing factories can pose a serious threat to public health. Biofilms are also potential reservoirs for antibiotic resistance genes and may contribute to their spread in the environment. Despite the use of systems to protect raw materials and products from contamination with potentially pathogenic microorganisms in food production plants, it is impossible to eliminate them. A significant problem that hinders their elimination is the ineffectiveness of the disinfection process and the growing resistance of bacteria to antibiotics. The presented results show that the tested *E. coli* strains are slightly better biofilm producers than the *S*. Senftenberg strains. The highest concentration of formaldehyde used completely inhibited the growth of most strains after 1 min of operation, while it inhibited all of them when the time of exposure to the disinfectant was extended to 15 min. It is worth emphasizing that 10-fold and 100-fold lower concentrations of formaldehyde had a disinfecting effect, significantly or even completely reducing the survival of the tested rods after 15 min of exposure. Thus, the antibiotic effect of formaldehyde was dependent not only on the concentration, but also on the duration of its action on bacteria. The obtained results are therefore important, because, due to the emergence of strains resistant to disinfectants that have been used for years, the use of these disinfectants is being abandoned and new ones are being sought. The use of formaldehyde and other chemicals as disinfectants requires awareness of both their benefits and the risks they can pose, especially concerning human health and the environment. Formaldehyde-based preparations, whose high antimicrobial effectiveness, even in low concentration ranges, results from the ability to affect both proteins and nucleic acids of microorganisms, are recommended for disinfecting heavily contaminated and hard-to-reach surfaces. Formaldehyde can be used for the disinfection of livestock buildings or other animal housing, disinfection of public premises, and the cleaning and disinfection of healthcare equipment, but only when exposure to humans and the environment is negligible. Therefore, it is important to educate factory and healthcare workers on the proper conduct of sanitary and hygienic processes that could potentially affect the safety of public health.

The results of our research indicate a good antibiofilm effect of most of the concentrations of the lingonberry extract used. However, the antibiofilm activity of the extract at higher concentrations was weaker for *S*. Senftenberg 132 and *E. coli* WW01. This interesting phenomenon is difficult to explain. One of the probable causes could be the hindered solubility of the active ingredients at the higher extract concentrations. Precipitation can result in a lower biofunctional availability of the extract. Also, while performing a broth microdilution assay, disorders in the turbidity of the precipitate may occur.

The composition of the lingonberry leaf extract used in our research has been described in our previous article [21]. The extract compounds were identified based on accurate mass searching, fragmentation analysis (MS/MS), comparison of accurate mass, and matching of the MS/MS pattern with standards and with data published in the literature. Flavonols (quercetin derivatives), phenolic acids (derivatives of caffeoylquinic, caffeoyl-hexose-hydro-xyphenol, and coumaroyl-hexose-hydroxyphenol acids), procyanidins (A and B dimmers), and iridoids were the most dominant compounds extracted from *V. vitis-idaea.* It is confirmed in the literature [48,49,50] that these substances exhibit antiadhesive and antibiofilm activities. The adhesion process is a very important step in the biofilm formation process. It was described that phenolics and flavanols present in plant extracts reduced bacterial adhesion [51]. The interaction of bioactive compounds with cell adhesion receptors can explain this phenomenon [52]. The antibiofilm activity of phenolics can also be connected with the rupture of the bacterium’s cytoplasmic membrane, reduction in membrane fluidity, and binding to bacterial DNA, leading to the inhibition of nucleic acids synthesis, cell wall synthesis, or energy metabolism [49,51]. Phenolic compounds are also known to be nonspecific inhibitors of autoinducer-mediated cell–cell signaling in bacteria. This effect can directly influence biofilm formation, since the communication between bacterial cells plays a crucial role in this process [45].

Both leaves and fruit extracts of *V. vitis-idaea* are of interest to researchers due to their antibacterial activities. However, their activity differs. The antimicrobial activity of berry extracts of *Vaccinium* spp. was shown against several Gram-negative (*Salmonella typhimurium*, *Pseudomonas fluorescens*, *Pseudomonas aeruginosa*, and *Serratia marcescens*) and Gram-positive (*Bacillus cereus*, *Staphylococcus aureus*, *B. subtilis*, and *Listeria monocytogenes*) bacteria [24,53,54]. The antibacterial effect of lingonberry fruits was studied against Gram-negative and Gram-positive bacteria [53]. Interestingly, the strongest antimicrobial effect was obtained against Gram-negative *Pseudomonas* spp. [49]. Kryvtsova et al. [24] showed that the leaf extract of *V. vitis-idaea* more effectively inhibited the process of biofilm formation by *S. aureus* strains than berry extract did. This difference may be related to their respective compositions. The leaf extract was characterized by a higher content of tannins and polyphenolics than the berry extract. Anthocyanins, vitamin C, and organic acids were found only in the berry extracts. Vernigorova and Buzuk [55] showed that the antibacterial activity of leaf extract is related to the presence of arbutin, methylarbutin, and ellagic acid. Fontaine et al. [56] established that ellagic acid inhibits biofilm formation, which may explain one of the mechanisms of the antibacterial activity of lingonberry leaf extract. It is also known that extraction methods play a crucial role in determining the concentration of active compounds in plant extracts. The choice of extraction method can significantly impact the yield and purity of the target compounds. Various extraction techniques (solvent extraction, Soxhlet extraction, supercritical fluid extraction (SFE), microwave-assisted extraction (MAE), ultrasound-assisted extraction (UAE), and pressurized liquid extraction (PLE)) are employed in scientific research, and each method has its advantages and limitations [57]. Moreover, it is suggested that plant materials originating from different geographic regions may differ in bioactivity, making it difficult to find a universal remedy to microbiological contamination.

## 4. Materials and Methods

### 4.1. Bacterial Strains

The study was carried out on five strains of *Salmonella enterica* subsp. *enterica* sv. Senftenberg (Table 3). Strains were isolated from poultry feed samples in 2014 at the LAB-VET Veterinary Diagnostic Laboratory (Tarnowo Podgórne, Poland) through procedures approved by the Polish Centre for Accreditation. They were serotyped at the National Serotype *Salmonella* Centre (Gdańsk, Poland). The strains were shown to be sensitive to ciprofloxacin (CIP, 5 μg), co-trimoxazole (STX, 25 μg), cefotaxime (CTX, 5 μg), amoxicillin/clavulanic acid (AMX 30; 20/10 μg), and ampicillin (AMP, 10 μg), as demonstrated in a previous publication [41]. The second group consists of six *E. coli* isolates derived from laying hens’ feces and cloacal swabs collected from a poultry farm in 2014 (Table 4). The birds were in the laying period—the time during which no antimicrobials were used. *E. coli* strains belonged to 4 different phylogenetic groups: A, B1, C, and D; one of them—the CM02 strain—was multi-drug-resistant (MDR), while the CJ01 isolate showed resistance to ampicillin [58]. *E. coli* isolates were stored at −80°C in TSB medium (Trypticase Soy Broth, Biomerieux, France) with an addition of 10% DMSO (Chempur, Piekary Śląskie, Poland) as a part of a collection of microorganisms at the Wrocław Medical University (Wroclaw, Poland). *Salmonella* strains came from a repository at the Faculty of Biological Sciences at the University of Wrocław (Wroclaw, Poland).

### 4.2. Chemical and Plant Material

Formaldehyde (37%) was purchased from Chempur (Piekary Śląskie, Poland) and dissolved in H_2_O_miliQ_ to obtain solutions 0.02%, 0.2%, and 2.0%. Dried leaves of *V. vitis-idaea* were purchased from “KAWON-HURT’’, a general partnership (Gostyń, Poland) with marketing authorization number IL-3333/LN.

### 4.3. Preparation of V. vitis-idaea Extract

The purchased dry leaves were ground into powder in an electric blender. A total of 20 g of leaf powder was dissolved in 180 mL of distilled water in a glass bottle, heated to 85 °C in a water bath, and kept at this temperature during shaking for 8 h. After cooling, the liquid was filtered through the Whatman No. 1 filter paper. The filtrate was then condensed and dried in a smaller glass bottle at 37 °C for 48 h. Then, the dried extract was dissolved in distilled water to obtain concentrations ranging from 0.005 to 2.0 mg/mL. Identification of lingonberry extract compounds was performed based on accurate mass searching, fragmentation analysis (MS/MS), comparison of accurate mass, and matching of MS/MS pattern with standards and with data published in the literature [22].

### 4.4. Biofilm Formation

The biofilm formation was evaluated in 96-well polystyrene U-bottom microplates according to the method described by O’Toole and Kolter [59] with minor modifications as described by Lamas et al. [60]. The experiment aimed to compare the intensity of biofilm formation by *Salmonellae* and *E. coli* strains. Wells of the microplates were filled with 100 μL of bacterial suspension (OD_550 nm_ 0.1, 0.5 in McFarland scale, densitometer, Biosan, Jozefow, Polska), prepared in TSB (Biomaxima, Lublin, Poland). Then, plates were incubated aerobically for 48 h at 37 °C. Each isolate was tested in triplicate. After incubation, the wells were emptied and washed twice with 200 μL of distilled water to remove free-floating cells. The bacteria that were attached to the walls were then fixed by adding 200 μL 96% methanol for 15 min. The plates were emptied and air-dried, and the wells were stained with 200 μL of 0.1% crystal violet solution (Sigma-Aldrich, Poznań, Poland) for 5 min. The excess crystal violet was depleted and washed three times with 200 μL of distilled water. The microplates were air-dried, and the dye that was bound to the adherent cells was resolubilized with 200 μL of 33% glacial acetic acid per well. Each well’s optical density (OD) was measured at 570 nm with a plate reader (ASYS UVM 340, Biochrom, Cambridge, UK). The OD was used to classify the isolates. Wells containing sterile TSB medium were recognized as a negative control.

### 4.5. Biofilm-Oriented Aseptic Test (BOAT)

In the presence of viable metabolically active bacteria, tetrazolium chloride (TTC) is reduced from a colorless compound to red formazan, which correlates to the number of viable cells, utilized in the BOAT method [61]. The strains were cultured in TSB (Biomaxima, Lublin, Poland) and incubated at 37 °C for 24 h. After incubation, the bacterial suspensions were standardized with PBS to reach a density of 0.5 McFarland scale (OD_550 nm_ 0.1, DEN-1B McFarland Densitometer, Biosan, Jozefów, Polska) and serially diluted to obtain 1 × 10^5^ CFU/mL. Subsequently, 5 × 100 μL of an individual bacterial strain’s suspension (1 × 10^5^ cells) was transferred to five adjacent wells of a 96-well polystyrene plate. This procedure was performed in duplicate (plate A and plate B). Next, the suspensions were incubated at 37 °C for 24 h. After 24 h, the suspensions from both plates were removed and thoroughly rinsed with 100 μL 0.9% NaCl. Next, (plate A) 100 μL of formaldehyde in concentrations of 2.0%, 0.2%, and 0.002% were added to the wells for selected contact times (1 and 15 min). After the contact time, the aseptic was removed and the wells were filled with a universal neutralizing agent (saline peptone water, Merck, Darmstadt, Germany) for 5 min. After this, the saline peptone water was removed. The wells were filled with 100 μL of TSB and with 5 μL of tetrazolium chloride (TTC) (Merck, Darmstadt, Germany), a reagent staining metabolically active microorganisms red. The results were assessed colorimetrically after 24 h of incubation of the plate at 37 °C. In the case of plate B, stages were performed in the same manner as in the case of plate A, however, working solutions of formaldehyde were not added (positive control). Plate B was used as a control of the strains’ ability to form biofilm. Negative control was also included in plate B, which had five wells containing sterile TSB. When reading the results on plates, the following criteria for negative results were assessed: no growth in five wells; no growth in four out of five wells; and no growth in three out of five wells. Positive results showing viable metabolically active bacteria were evidenced by the growth of microorganisms in five wells, growth in four out of five wells, and growth in three out of five wells.

### 4.6. Quantitative Measurements of the Number of Colony-Forming Units in BOAT Assay

Since *Salmonellae* and *E. coli* strains were incubated in polystyrene plates, viable metabolically active bacteria reduced TTC into red formazan. The excess bacterial suspension was depleted and washed three times with 200 μL of distilled water. To count biofilm-forming cells, 100 μL of mild detergent 0.5% Triton X-100 was used twice to detach cells from the walls of the wells. Then, the solution was transferred to 0.9% NaCl to perform serial dilutions. Next, the bacterial suspensions were spread on the agar plates and incubated at 37 °C. After 24 h of incubation, the number of CFU/mL was counted.

### 4.7. Antibiofilm Properties of Lingonberry (V. vitis-idaea) Extract 

This experiment aimed to assess the influence of lingonberry extract on biofilm formation. Wells of the microplates were filled with 100 μL of bacterial suspension (OD_550 nm_ 0.1, 0.5 in McFarland scale, densitometer, Biosan, Jozefów, Poland) prepared in TSB (Biomaxima, Lublin, Poland). Previously prepared dilutions of the extract (100; 50; 10; 1; 0.5; 0.25 mg/mL) at the volume of 2 μL were added to the wells, where they reached relevant concentrations of 2; 1; 0.2; 0.02; 0.01; and 0.005 mg/mL. After 48 h of incubation at 37 °C, quantitation of biofilms was measured as described in biofilm formation assay (Section 4.4). The wells with bacteria incubated in TSB were regarded as a positive control. Wells containing sterile TSB medium were recognized as a negative control.

### 4.8. Statistical Analysis

The nonparametric Mann–Whitney U test was used in the analysis of the results determining the effect of formaldehyde, and the parametric Student’s *t*-test was used for independent samples in the analysis of the results determining the effect of the lingonberry extract on the tested bacterial strains. Statistical calculations were performed using Statistica 13.3 (Stat Soft, Krakow, Poland). All values are expressed as mean ± SD. The experiments were repeated three times. Values of *p* ≤ 0.05 were considered statistically significant.

## 5. Conclusions

The overuse of antibiotics and other chemical treatments has increased the emergence of antibiotic-resistant bacteria [62]. The increase in the incidence of food-borne diseases requires new and effective methods to reduce the contamination of food with pathogenic microorganisms [63]. Currently, there is a noticeable trend of withdrawing formaldehyde from many industries due to its toxic properties, but the idea of replacing chemicals with natural products is still a big challenge. It has been shown that the final effect of the bactericidal activity of formaldehyde and lingonberry extract depended on the strain. Within a certain formaldehyde concentration range (0.2 and 2.0%), a significant reduction in the number of viable cells in biofilms was possible. However, in lower concentrations (0.02%), one cannot observe any changes in bacterial survival, even when extending the exposure time. Plant extracts may also be insufficient for the effective elimination of bacteria existing in consortia or, as desired, they may reduce biofilm formation; however, finding a universal plant extract that works on bacteria globally can be very difficult.

## Figures and Tables

**Figure 1 ijms-24-14579-f001:**
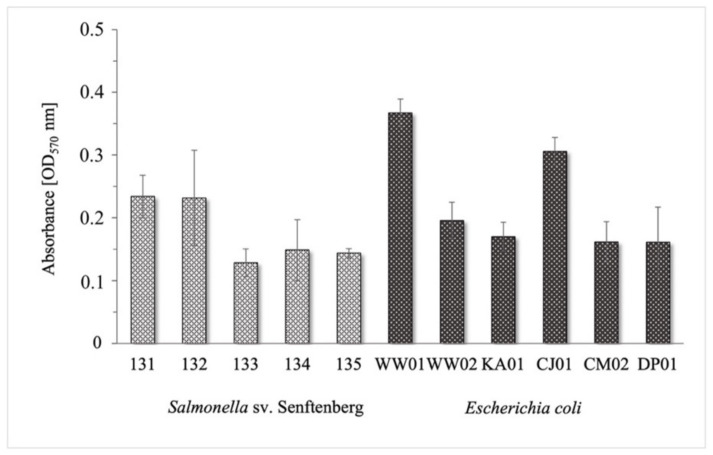
Graph showing the optical density (OD) of biofilm formation by *Salmonella* Senftenberg and *E. coli* strains.

**Figure 2 ijms-24-14579-f002:**
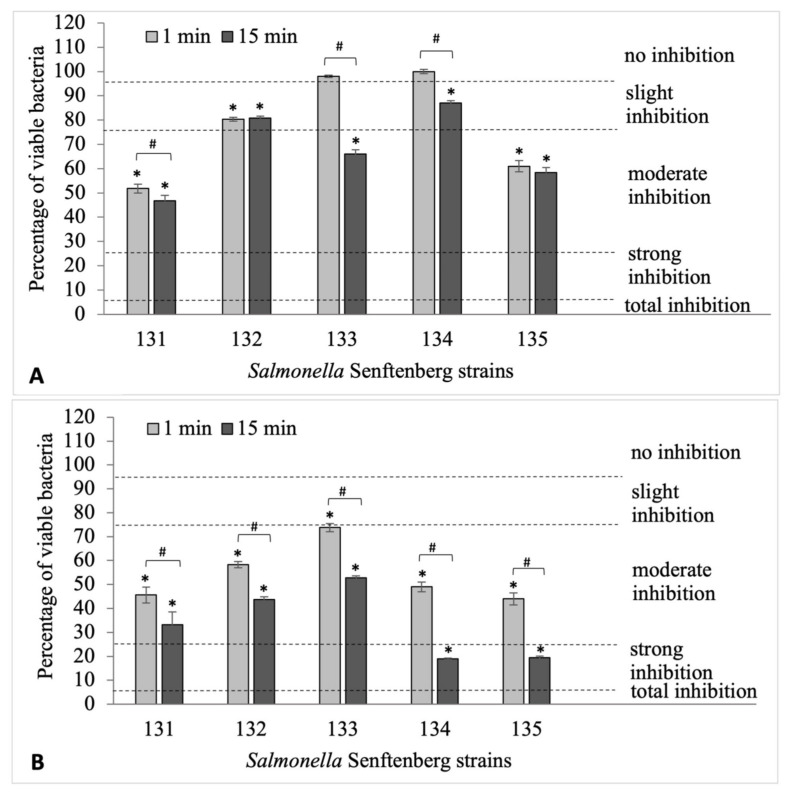
The viability of *S.* Senftenberg strains in biofilms after the application of formaldehyde at the concentrations of 0.02% (**A**) and 0.2% (**B**) for 1 and 15 min. Statistically significant results (*p* ≤ 0.05) of the tested samples compared with the control samples are marked with an asterisk (*); statistically significant differences (*p* ≤ 0.05) between samples treated with formaldehyde for 1 min and 15 min were noted with a hashtag (#). The experiment was repeated three times.

**Figure 3 ijms-24-14579-f003:**
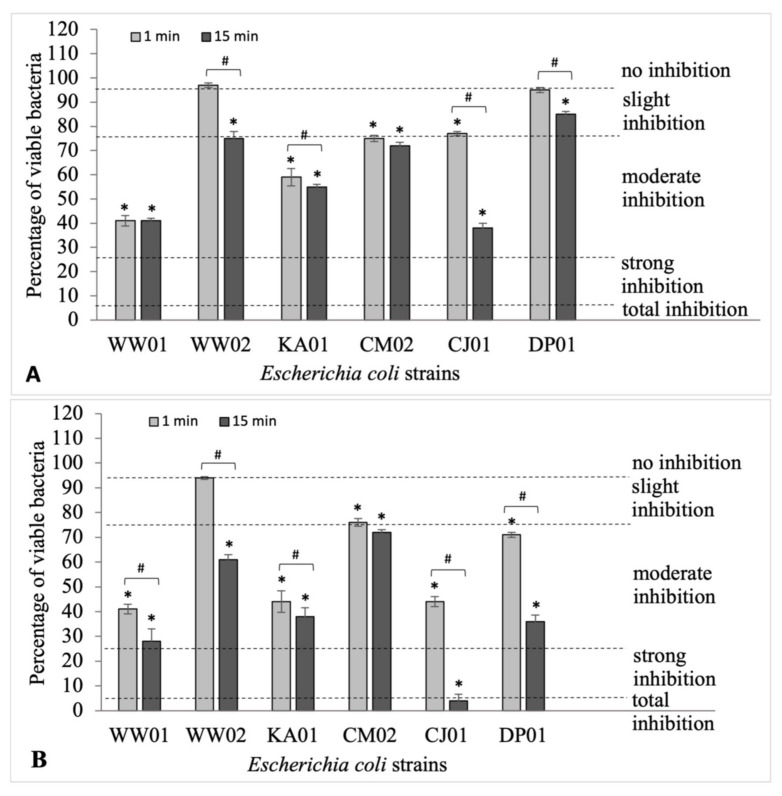
The viability of *E. coli* strains in biofilms after the application of formaldehyde at the concentrations of 0.02% (**A**) and 0.2% (**B**) for 1 and 15 min. Statistically significant results (*p* ≤ 0.05) of the tested samples compared with the control samples are marked with an asterisk (*); statistically significant differences (*p* ≤ 0.05) between samples treated with formaldehyde for 1 min and 15 min were noted with a hashtag (#). The experiment was repeated three times.

**Figure 4 ijms-24-14579-f004:**
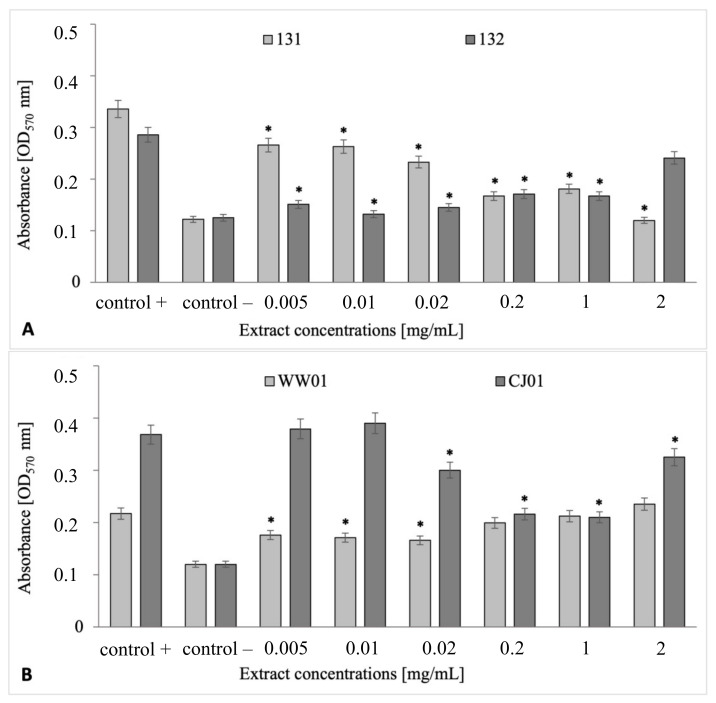
Antibiofilm properties of *V. vitis-idaea* extract in biofilms formed by *S*. Senftenberg 131 and 132 (**A**) and *E. coli* WW01 and CJ01 (**B**) strains. Statistically significant results (*p* ≤ 0.05) of the tested samples compared with the control samples are marked with an asterisk (*).

**Table 1 ijms-24-14579-t001:** Antibiofilm effect of formaldehyde against *Salmonella* Senftenberg using the BOAT method.

Strain No.	131		132		133		134		135	
	Incubation Time (Min)									
**Formaldehyde concentration (%)**	1	15	1	15	1	15	1	15	1	15
0.0 (positive control)	+	+	+	+	+	+	+	+	+	+
0.02	+	+	+	+	+	+	+	+	+	+
0.2	+	+	+	+	+	+	+	+	+	+
2.0	−	−	−	−	−	−	−	−	−	−

“+” incomplete eradication of biofilm (reduction of colorless 2,3,5-triphenyltetrazolium chloride (TTC) to red 1,3,5-triphenylformazan (TPF), which indicated the presence of metabolically active bacteria); “−” complete eradication of biofilm (lack of TTC reduction, which indicated the inhibition of the metabolic activity of the bacteria); positive control—samples without formaldehyde.

**Table 2 ijms-24-14579-t002:** Antibiofilm effect of formaldehyde against *E. coli* strains using the BOAT method.

Strain No.	WW01		WW02		KA01		CM02		CJ01		DP01	
	Incubation Time (Min)											
**Formaldehyde concentration (%)**	1	15	1	15	1	15	1	15	1	15	1	15
0.0 (positive control)	+	+	+	+	+	+	+	+	+	+	+	+
0.02	+	+	+	+	+	+	+	+	+	+	+	+
0.2	+	+	+	+	+	+	+	+	+	+	+	+
2.0	+	−	−	−	−	−	−	−	+	−	−	−

“+” incomplete eradication of biofilm (reduction of colorless 2,3,5-triphenyltetrazolium chloride (TTC) to red 1,3,5-triphenylformazan (TPF), which indicated the presence of metabolically active bacteria); “−” complete eradication of biofilm (lack of TTC reduction, which indicated the inhibition of the metabolic activity of the bacteria); positive control—samples without formaldehyde.

**Table 3 ijms-24-14579-t003:** *Salmonella* Senftenberg strains used in the experiments.

Species	Subspecies	Serovar	Internal Collection No.	Antibiotic Resistance [41]
*Salmonella enterica*	*enterica*	Senftenberg	131	not found
	*enterica*	Senftenberg	132	not found
	*enterica*	Senftenberg	133	not found
	*enterica*	Senftenberg	134	not found
	*enterica*	Senftenberg	135	not found

**Table 4 ijms-24-14579-t004:** *E. coli* strains used in the experiments.

*E. coli* Strain	Phylogenetic Group	Antibiotic Resistance [58]
KAO1	A	not found
CJ01	B1	not found
DP01	C	not found
CM02	D	AMP/TET/SXT/PRL
WW01	B1	AMP
WW02	A	not found

Abbreviations: AMP—ampicillin (10 μg), TET—tetracycline (10 μg), SXT—trimethoprim/sulfamethoxazole (1.25/23.75 μg), PRL—piperacillin (100 μg).

## Data Availability

The data presented in this study are available on request from the corresponding author.

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
