# Peer review of "Exposing Salmonella Senftenberg and Escherichia coli Strains Isolated from Poultry Farms to Formaldehyde and Lingonberry Extract at Low Concentrations"

_ijms, 2023, doi:10.3390/ijms241914579_

Round 1
Reviewer 1 Report
In this research paper, an evaluation was conducted to assess the antibiofilm properties of two substances: formaldehyde and Lingonberry extract. The experimental strains used in the study were Salmonella and Escherichia coli. The results revealed that formaldehyde demonstrated an effective capability to eradicate bacterial cells of both species within the biofilm. On the other hand, Lingonberry extract exhibited a degree of anti-adhesive activity specifically against S. senftenberg.
Minor comments:
Line 221-224, how do exponential trend lines represent the relationship between concentration of lingonberry extract and adhesion of bacteria? Could author explain the mean of R2 values in Figure 4?
Author Response
Reviewer 1
Thank you for taking up the revision process. We are happy that you have had an independent opinion on our manuscript.
We have decided to remove the R2 values in Figure 4. In theory, R2 value is an indicator of how closely the data fit the trendline. R-squared is a number between 0 and 1. The higher the R-squared value, the better the model explains the variability in the data. In practice, this value can help determine whether a model is suitable for describing a given phenomenon or whether it can be used to predict future values. Values below 0.5 should not be indicated in the analysis, hence we have removed them from charts A and B in Figure 4. Thank for very much for drawing attention to it, honestly, that was our oversight. In connection with that we have also removed the sentence linked to the R2: Creating the exponential trend lines helped to observe the relationship between the increasing concentration (in the range of 0.005–2 mg/mL) of lingonberry extract and the adhesion ability of bacteria to the microtiter plate surface (Figure 4).
Reviewer 2 Report
Formaldehyde information is inconsistent. In the introduction, the authors write:
Line 52 It is worrying that the number of reports of poultry products contaminated with Salmonella bacteria has increased in recent years. This is partially due to the prohibition of formaldehyde usage to disinfect poultry and pig feed in the European Union.
Line 82 A procedure often used in poultry farming is disinfection breeding rooms with the use of potassium permanganate and formaldehyde solutions.
Formaldehyde cannot be used to disinfect poultry and pig feed, but can it be used to disinfect breeding rooms?
In the discussion:
Line 241-242 Literature data focuses on studies of formaldehyde as an agent used at stages of feed processing, as well as a feed additive. Formaldehyde is also used to disinfect rooms and devices used in industrial units, which effectively reduces the risk of food contamination with Salmonella rods [32,33].
Please pay attention to the date of publication of the cited research and the year when prohibition on formaldehyde. In which countries were these studies carried out?
Please clarify this informations.
Please explain why in "Antiadhesive Properties of Lingonberry (V. vitis-idaea) Extract" and "Biofilm formation" analysis biofilm was tested after 48 h but in "Biofilm-oriented Aseptic Test (BOAT)" biofilm was tested after 24 h.
Please explain the difference between the methodology "4.7. Antiadhesive Properties of Lingonberry (V. vitis-idaea) Extract" and "4.4. Biofilm Formation". In my opinion, these are identical procedures, but the authors calls them differently.
The title of this work is "Anti-biofilm Effectiveness of Low Concentrations of Formaldehyde and Lingonberry Extract". In my opinion, the aspect about the extract, which I think is the most interesting part, has been partially omitted.
The anti-biofilm properties of Lingonberry Extract have not been analysed. The analysis of Biofilms Eradication Effectiveness was performed only for Formaldehyde. Why?
Author Response
Reviewer 2
Thank you for your valuable engagement in reviewing the manuscript. Below we have listed answers to your suggestions:
In Introduction:
Thank you for finding this inconsistency (lines 52 and 82) about formaldehyde usage, in the Introduction section. We have introduced tiny corrections in lines 52-53 as well as removed sentences between lines 81-83.
In the Discussion section:
We have removed the sentences between lines 240-242: Literature data focuses on studies of formaldehyde as an agent used at stages of feed processing, as well as a feed additive. Formaldehyde is also used to disinfect rooms and devices used in industrial units, which effectively reduces the risk of food contamination with Salmonella rods [32,33].
In turn, below (lines 247-248), we have added information related to the topic review paper about formaldehyde employment (cited in the previous version of the manuscript): “Poultry application and antibacterial mechanisms were extensively described in review by Ricke et al. [33].”
We have clarified further inconsistencies:
1st suggestion: Please explain why in "Antiadhesive Properties of Lingonberry (V. vitis-idaea) Extract" and "Biofilm formation" analysis biofilm was tested after 48 h but in "Biofilm-oriented Aseptic Test (BOAT)" biofilm was tested after 24 h.
Answer: Biofilm formation assay was optimized for the tested strains. In this work, we have shown the results obtained after 48h, when the differences between strains could be observed. We have also performed experiments where the incubation time was 24 h, but the amount of biofilm was not sufficient (too thin, too less) to compare strains (coloration similar to the background signal). In the antiadhesive test, we followed the same manner of thinking. Moreover, we tried to refer to the methods in the articles cited to make it easy to compare the results across laboratories. In conclusion, in all three assays, we were trying to use the lowest time of incubation to observe significant differences between bacterial isolates.
2nd suggestion: Please explain the difference between the methodology "4.7. Antiadhesive Properties of Lingonberry (V. vitis-idaea) Extract" and "4.4. Biofilm Formation". In my opinion, these are identical procedures, but the authors call them different.
Answer: These two methods look similar, but in fact there are some differences. To clarify the main differences, we have reedited method 4.7. referring to 4.4.
3rd suggestion: The title of this work is "Anti-biofilm Effectiveness of low concentrations of Formaldehyde and Lingonberry Extract". In my opinion, the aspect about the extract, which think is the most interesting part has been partially omitted.
Answer: We have thought about this issue and finally proposed another one: “Exposure of SalmonellaSenftenberg and Escherichia coli Strains Isolated from Poultry Farms to Formaldehyde and Lingonberry Extract Used in Low Concentrations”. We believe it better reflects the content of this work.
4th question: The anti-biofilm properties of Lingonberry Extract have not been analysed. The analysis of Biofilms Eradication Effectiveness was performed only for Formaldehyde. Why?
Answer: Lingonberry Extract was tested in the previous publication, where authors found out that extract in high concentration (10.0, 15.0, 20.0 mg/mL) was effective in the reduction of E. coli planktonic growth. The effect of plant extract was investigated on biofilm formation too. The authors noticed that statistically significant inhibition of biofilm-mass formation in the presence of extract at the concentration of 0.125 mg/mL was noticed after 4, 5, 6, and 10 days of bacterial incubation (Wojnicz et al., see below). Moreover, in our opinion, the performance of the BOAT method with lingonberry extract wouldn’t be novel in this research. Anti-biofilm activity of the V. vitis-idaea extracts has been extensively studied by other authors, whereas anti-adhesive effectiveness is still not fully understood.
Wojnicz D, Kucharska AZ, Sokół-Łętowska A, Kicia M, Tichaczek-Goska D. Medicinal plants extracts affect virulence factors expression and biofilm formation by the uropathogenic Escherichia coli. Urol Res. 2012, Dec;40(6):683-97. doi: 10.1007/s00240-012-0499-6. Epub 2012 Aug 23. PMID: 22915095; PMCID: PMC3495101.
Round 2
Reviewer 2 Report
Thank you for your answer. The authors responded to all my suggestions.